# Lookback for Learning to Branch

**Prateek Gupta,** *University of Oxford, The Alan Turing Institute*      *pgupta@robots.ox.ac.uk*
**Elias B. Khalil,** *University of Toronto*      *khalil@mie.toronto.edu*
**Didier Chételat,** *CERC, Polytechnique Montréal*      *didier.chetelat@polymtl.ca*
**Maxime Gasse,** *Mila, CERC, Polytechnique Montréal*      *maxime.gasse@polymtl.ca*
**Yoshua Bengio,** *Mila, Université de Montréal, CIFAR Fellow*      *yoshua.bengio@mila.quebec*
**Andrea Lodi,** *CERC, Polytechnique Montréal, and Cornell Tech and Technion - IIT*      *andrea.lodi@cornell.edu*
**M. Pawan Kumar,** *University of Oxford*      *pawan@robots.ox.ac.uk*

**Reviewed on OpenReview:** *https://openreview.net/forum?id=EQpGkw5rvL*

## Abstract

The expressive and computationally efficient bipartite Graph Neural Networks (GNN) have been shown to be an important component of deep learning enhanced Mixed-Integer Linear Programming (MILP) solvers. Recent works have demonstrated the effectiveness of such GNNs in replacing the *branching (variable selection) heuristic* in branch-and-bound (B&B) solvers. These GNNs are trained, offline and on a collection of MILP instances, to imitate a very good but computationally expensive branching heuristic, *strong branching*. Given that B&B results in a tree of sub-MILPs, we ask (a) whether there are strong dependencies exhibited by the target heuristic among the neighboring nodes of the B&B tree, and (b) if so, whether we can incorporate them in our training procedure. Specifically, we find that with the strong branching heuristic, a child node's best choice was often the parent's second-best choice. We call this the "lookback" phenomenon. Surprisingly, the branching GNN of Gasse et al. (2019) often misses this simple "answer". To imitate the target behavior more closely, we propose two methods that incorporate the lookback phenomenon into GNN training: (a) target smoothing for the standard cross-entropy loss function, and (b) adding a *Parent-as-Target (PAT) Lookback* regularizer term. Finally, we propose a model selection framework that directly considers harder-to-formulate objectives such as solving time. Through extensive experimentation on standard benchmark instances, we show that our proposal leads to decreases of up to 22% in the size of the B&B tree and up to 15% in the solving times.

## 1 Introduction

Many real-world decision making problems are naturally formulated as Mixed-Integer Linear Programs (MILPs), for example, process integration (Kantor et al., 2020), production planning (Pochet & Wolsey, 2006), urban traffic management (Foster & Ryan, 1976; Fayazi & Vahidi, 2018), data center resource management (Nowatzki et al., 2013), and auction design (Abrache et al., 2007), to name a few. In some applications, such as urban traffic management (Fayazi & Vahidi, 2018), these MILPs need to be solved frequently (e.g., every second) with only a slight change in the specifications. Similarly in data center resource management (Nowatzki et al., 2013), the available machines and tasks that must be served evolve over time, prompting repeated assignments through MILP solving. Even with a linear objective function and linear constraints, the requirement that some decision variables must take on integer values makes MILPs NP-Hard (Papadimitriou & Steiglitz, 1982).

As a result, the Branch-and-Bound (B&B) algorithm (Land & Doig, 1960) is used in the modern solvers to effectively prune the search space of the MILPs to find the global optimum. B&B proceeds by recursively splitting the search space and solving the linear relaxation of the resulting sub-problems, the solution of

which serves as an informative bound to prune the search space. The algorithm continues until a solution with integral decision variables is found and proven optimal. Quite naturally, the sub-problems resulting from the branching decisions at each step of the algorithm can be represented as a tree; every node (a sub-MILP) has a parent except the root node.

While seemingly simple, the B&B algorithm must repeatedly make search decisions that are crucial for its efficiency, such as the choice of decision variable over which to branch at each iteration, a problem known as *variable selection*. Even though the worst-case time-complexity of the B&B algorithm is exponential in the size of the problem (Wolsey, 1988), it has been successfully applied in practical settings thanks to the careful design of a number of effective search heuristics.

Modern solvers are configured with expert-designed heuristics and are aimed at solving general MILPs. However, assuming that MILPs come from a specific distribution, there has been a recent surge in research related to statistical approaches to learning such heuristics, e.g., (He et al., 2014b; Alvarez et al., 2017; Khalil et al., 2016; Gasse et al., 2019; Zarpellon et al., 2020; Gupta et al., 2020; Huang et al., 2022).

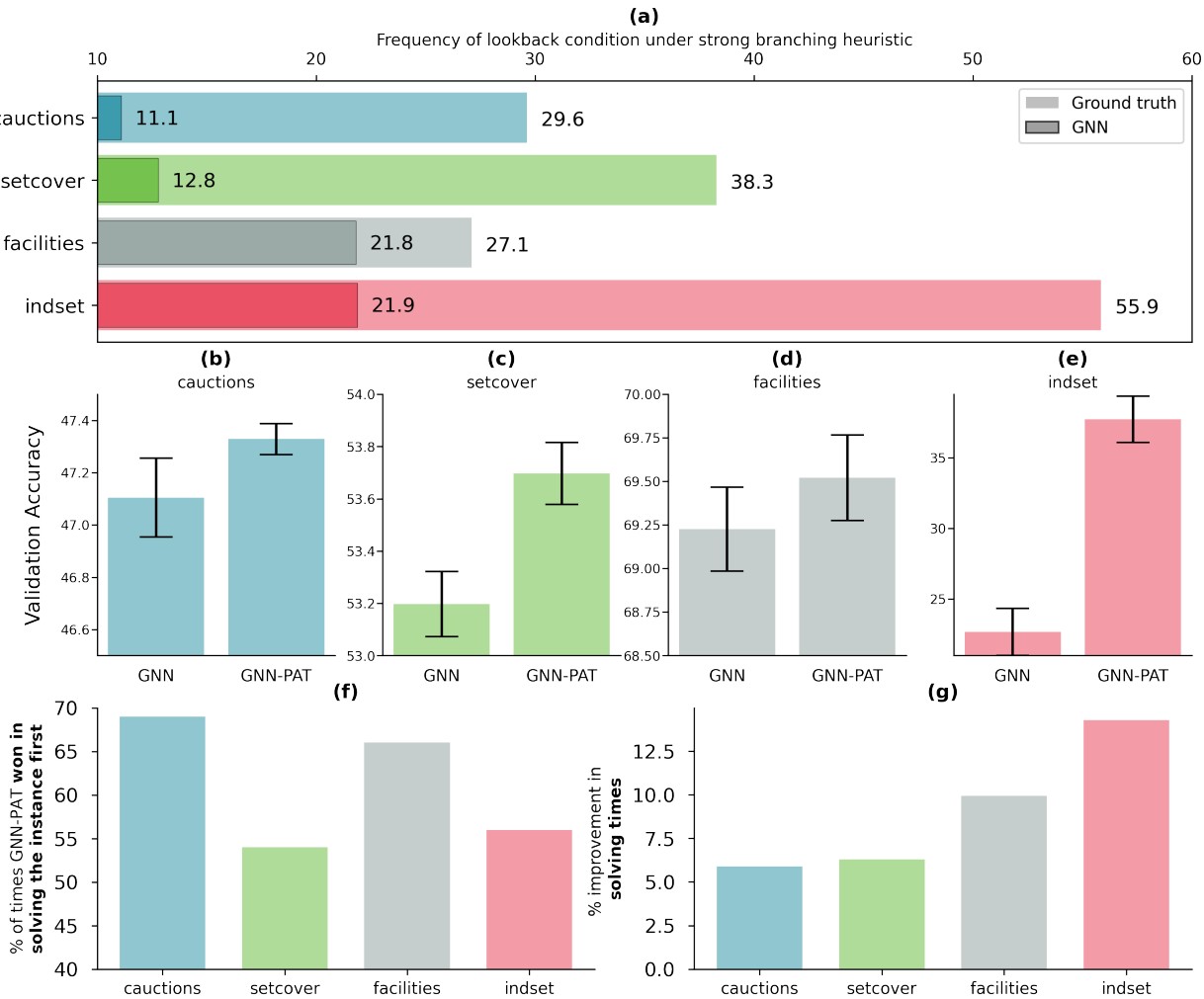

Figure 1: **(a)** Frequency of the "lookback" property when (i) instances are solved using the strong branching heuristic (Ground truth), and (ii) corresponding frequency with which GNNs would have respected this property on Ground truth. **(b)**-**(e)** GNNs trained with the proposed techniques (GNN-PAT) have higher accuracy on the validation dataset. **(f)** GNN-PAT solves test instances faster than other baselines resulting in **(g)** less time as compared to GNNs.

Gasse et al. (2019) proposed to use Graph Neural Network (GNN) for the variable selection problem to imitate the branching decisions of a computationally expensive *strong branching* heuristic that yields the smallest B&B trees in practice on a number of benchmarks. The GNN operates on a bipartite representation of a MILP in which variables and constraints are nodes and an edge indicates that a variable has a non-zero coefficient in a constraint. Such a bipartite representation has several advantages. First, it is able to capture the most crucial invariances of MILPs, namely, permutation invariance to the ordering of decision variables/constraints, and, by way of feature engineering, scale invariance to the scaling of constraints or objective coefficients. Second, due to the shared parametric representation, the model can be applied to MILPs of arbitrary size. Thus, GNNs have been extensively used to process MILPs as inputs to a neural network aimed at imitating various superior heuristic decisions (Huang et al., 2022; Zarpellon et al., 2020; Nair et al., 2020; Khalil et al., 2022), often outperforming off-the-shelf solvers with their default parameters/heuristics.

The training of GNNs consists primarily of two steps: (a) *Dataset collection*: data, i.e., sub-problems and corresponding strong branching decisions, are collected offline by solving reasonably-sized MILPs while invoking, with some exploration probability, either a less efficient but faster heuristic, or the strong branching heuristic, and (b) *Model Training*: assuming independent and identical distribution (i.i.d.) of the collected dataset, GNNs are trained to minimize standard cross-entropy loss between the predictions of GNNs and the strong branching target.

So far, previous works on learning to branch have tried imitating the strong branching decisions at each node in isolation, thereby ignoring the statistical resemblance in the behavior of the neighboring nodes. However, it often happens that strong branching's top choice at a node was the *second-best* choice at the node's parent, a phenomenon we will refer to onwards as the "lookback" property of strong branching. Figure 1(a) shows how frequently this occurs in some standard benchmark instances, labeled for convenience as cauctions (Combinatorial Auction), setcover (Minimum Set Cover), facilities (Capacitated Facility Location), and indset (Maximum Independent Set); see Appendix A for similar statistics on some real-world instance sets, where the lookback property is equally prevalent. Given the importance of decisions closer to the root node, Figure 2 further shows that this phenomenon is quite prevalent at the top of the tree. Although it appears quite common empirically, this property has never been explicitly pointed out in the literature.

Since imitation learning aims to mimic the expert as closely as possible, and the expert exhibits this lookback property, it follows that successful imitation should exhibit the lookback property as well. However, as can be seen in Figures 1(a) and 2, there is a big gap between how frequently vanilla GNNs respect the lookback condition and the fraction of times it is actually observed. Towards closing this gap, we propose two approaches that encourage GNNs trained by imitation learning to respect the lookback condition, and analyze the effects on solving performance.

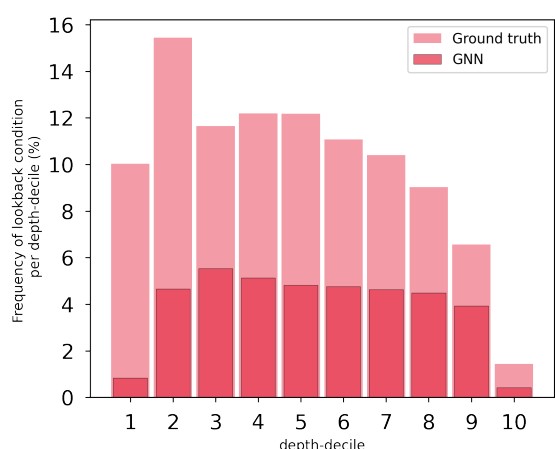

Figure 2: (Maximum Independent Set) Frequency of the lookback property per depth-decile, one of the 10 equally divided portions by depth of the B&B tree. Ground truth is obtained by solving small instances using strong branching heuristic. Retrospectively, GNNs do not respect the lookback property well enough. For similar statistics on other benchmark problem families, see Appendix B.

A difficulty is that our proposed methods introduce new hyperparameters, which raises the question of how to select them efficiently. Gupta et al. (2020) used validation accuracy on a few thousand observations collected by solving MILPs of a slightly bigger size than the training instances. We argue that such a selection strategy may not serve the true objectives that a practitioner might have. For instance, they might prefer a learning-enhanced solver with as small a running time as possible, regardless of the validation accuracy relative to the strong branching oracle. Thus, we also design an improved model selection framework to respect such varied requirements. As shown in Figure 1(b)-(e), the resulting combined method (GNN-PAT)

not only increases the validation accuracy on each benchmark, but also leads to decreases in the size of the final B&B tree by up to 22% and in the running time by up to 15% (Figure 1(g)), solving most of the instances fastest among the baselines (Figure 1(f)).

To summarize, the contributions of this paper are as follows.

- First, we demonstrate through numerical experiments that the "lookback" phenomenon often occurs in practice (see Figure 1 and 2).

- Second, we propose two ways of exploiting this phenomenon, namely through target softening, and through a regularizer term that encourages the GNN to exhibit this phenomenon.

- Third, we propose a model selection framework to incorporate harder-to-formulate practical objectives such as minimum solving times.

The paper is divided as follows. Sections 2 and 3 review the literature and preliminary notation and definitions, respectively. Section 4 proposes techniques to exploit the lookback phenomenon in imitation learning. Section 5 details our proposed hyperparameter selection framework. Then, Section 6 presents experimental results that show the benefits of these adjustments. Finally, in Section 7, we discuss implications of our proposals, limitations, and potential future directions, concluding in Section 8.

## 2 Related Work

The problem of variable selection in B&B based MILP solving has been studied quite extensively. While the gold standard strong branching heuristic yields the smallest B&B trees, due to the high computational cost per iteration it is impractical (see Section 3 for details). Thus, in its early years, the focus of research has been on hand-designed heuristic methods (Applegate et al., 1995; Linderoth & Savelsbergh, 1999) that are faster and sufficiently good. As a result, *reliability pseudocost branching* (RPB) (Achterberg et al., 2005) became the preferred branching strategy to solve general MILPs. RPB combines the lookahead strengths of strong branching heuristic with faster estimations offered through computationally inexpensive measures of performance (such as *pseudocosts* (Linderoth & Savelsbergh, 1999)).

In the last decade, researchers have proposed machine learning (ML) methods to imitate the strong branching heuristic, thereby leveraging the computationally inexpensive nature of the learned functions. For example, Alvarez et al. (2017) used extremely randomized trees on the data collected offline, whereas Khalil et al. (2016) used support vector machines to imitate the strong branching ranking from the first few hundred nodes explored in the B&B tree. We refer the reader to Lodi & Zarpellon (2017) for a detailed survey on ML-based branching strategies.

Both Alvarez et al. (2017) and Khalil et al. (2016) learn a classifier on fixed-dimensional, hand-designed features. Recently, however, several works have proposed deep-learning based branching strategies, starting from (Gasse et al., 2019). Their GNN approach has been the basis of several following work. Specifically, Gupta et al. (2020) explored the relation between the original MILP and subsequent sub-MILPs in a B&B tree to design a CPU-efficient neural network architecture. However, this relationship has more to do with the B&B characteristics than the oracle behavior. Similarly, Zarpellon et al. (2020) explored learning suitable representations based on the evolution of the B&B tree. Peng & Liao (2022) investigated, in the context of combinatorial auctions instances only, the effects of B&B node sampling to collect the training data. Finally, Nair et al. (2020) proposed a GPU-friendly alternating descent method of multipliers approach to solving linear programs that could be use to run strong branching on larger instances, allowing them to show that the approach could scale to large, heteroneous datasets. In none of these works, however, has the parent-child lookback property been explored in the context of training machine learning based variable selection strategies.

More recently, works have tried to move beyond imitation learning and have investigated reinforcement learning formulations for learning to branch. Sun et al. (2020) used a simple evolutionary strategies approach, but they only obtained improvements on very homogeneous benchmarks, while reporting subpar results

on the harder benchmarks of Gasse et al. (2019) used in this work. Zhang et al. (2022) used imitation learning framework of Gasse et al. (2019) to pre-train GNNs before using a hybrid reinforcement learning and Monte-Carlo tree search framework, reporting more encouraging results. In parallel, Etheve et al. (2020) proposed a Q-learning approach on value functions representing subtree size, and Scavuzzo et al. (2022) reinterpreted their approach as reinforcement learning on a tree-shaped Markov decision processes. Parsonson et al. (2022) proposed a similar mechanism, where the search tree is divided into small diving paths, that are used as imitation targets. These works all seek to address the problem of the long episode lengths using topological information from the branch-and-bound tree.

Although an interesting step forward, these approaches are nonetheless not currently competitive with imitation learning methods, which remain the state of the art. Indeed, they face unusual challenges in this context, including that poor decisions lead to longer, rather than shorter episodes, and also that transitions between states are also particularly computationally slow, since they involve solving linear programs. For these reasons, like most works, we chose to focus on the more successful strategy of imitation learning. Nonetheless, it is plausible that encouraging a lookback property in a reinforcement learning policy could lead to similar benefits to those explored in the context of this work.

## 3  Preliminaries

A MILP is a mathematical optimization problem characterized by a linear objective function and linear constraints in variables $\mathbf{x}$. A generic representation of a MILP is as follows:

$$\min_{\mathbf{x}} \mathbf{c}^\mathsf{T}\mathbf{x}, \quad \text{s.t.} \quad \mathbf{A}\mathbf{x} \leq \mathbf{b}, \quad \mathbf{x} \in \mathbb{Z}^p \times \mathbb{R}^{n-p}, \tag{1}$$

where $\mathbf{c} \in \mathbb{R}^n$ is the vector of cost coefficients, $\mathbf{A} \in \mathbb{R}^{m \times n}$ is the matrix of constraint coefficients, $\mathbf{b} \in \mathbb{R}^m$ is a vector of constant terms of constraints, and $p > 0$ decision variables are constrained to take integer values.

The B&B algorithm proceeds as follows. The Linear Programming (LP) relaxation of the MILP, obtained by relaxing the integer constraints on the discrete variables, is solved to obtain a lower bound on the global optimum, thereby resulting in $\mathbf{x}^*$ as the optimal solution. If $\mathbf{x}^*$ has integral values for the integer-constrained decision variables, then it is integer-optimal and the algorithm terminates. If not, one of the decision variables with fractional value, $k \in \mathcal{C}$, such that $\mathcal{C} \in \{k \mid \mathbf{x}_k^* \notin \mathbb{Z}, k \leq p\}$, is selected to split the MILP in two sub-MILPs. The resulting sub-MILPs are obtained by adding additional constraints $x_k \leq \lfloor x_k^\star \rfloor$ and $x_k \geq \lceil x_k^\star \rceil$, respectively. We denote as $\mathcal{C}$ the set of *branching candidates*, while the variable $x_i$ is termed as a *branching variable*. The algorithm proceeds recursively in this fashion by selecting the next sub-MILPs to operate on.

Denoting the optimal value of Eq. (1) by $P$, and using the superscripts 0 to denote the parent MILP, $-$ to denote the child MILP obtained by adding the lower bound to the branching variable, and $+$ otherwise. The strong branching heuristic selects the variable $x_{sb}$ that has the maximum potential to improve the bound, namely

$$x_{sb} = \arg\max_{k \in \mathcal{C}} \left[ \max\{P_k^- - P^0, \epsilon_{LP}\} \cdot \max\{P_k^+ - P^0, \epsilon_{LP}\} \right],$$

where $\epsilon_{LP} > 0$ is a small enough value to prevent the scores to collapse to 0 because of no improvement on either side of the branching (Achterberg, 2007).

## 4  Methodology

In this section, we describe our proposals to incorporate dependencies between successive nodes. A bipartite graph representation of a MILP is denoted by $\mathcal{G} \in (\mathbf{V}, \mathbf{E}, \mathbf{C})$, where $\mathbf{V} \in \mathcal{R}^{n \times d_v}$, $\mathbf{E} \in \mathcal{R}^{k \times d_e}$, and $\mathbf{C} \in \mathcal{R}^{m \times d_c}$. Here, $\mathbf{V}$ is the matrix of features for $n$ decision variables, $\mathbf{C}$ is the matrix of features for $m$ constraints, and $\mathbf{E}$ is the matrix of features for $k$ variable-constraint pairs. The terms $d_v$, $d_c$, and $d_e$ are the numbers of input features.

The data is collected by using the strong branching heuristic to solve instances of manageable size, thereby yielding $N$ graphical representations of MILPs, $\{\mathcal{G}_i\}_{i=1}^N$, the set of candidate decision variables, $\mathcal{C}_i$, with fractional value at a node $i$, and their corresponding strong branching scores $\mathbf{s}_i \in \mathbb{R}_{\geq 0}^{|\mathcal{C}_i|}$. We use $\mathbf{s}_{i,j}$ to denote the strong branching score of the $j^{th}$ candidate in $\mathcal{C}_i$. We denote the strong branching target chosen during the solving procedure by $y_i$, and the set of second-best strong branching variables by $\mathcal{Z}_i = \arg\max_{j \neq y_i} \mathbf{s}_{i,j}$. Thus, $\mathcal{Z}_i$ may include variables which have the same strong branching score as $y_i$ if there is a tie, else it includes all the variables with the second-best strong branching score, which can be more than one.

We denote the parent graph by a superscript 0 and the child node by a superscript 1. Thus, the parent bipartite graph of the $i$-th observation is denoted by $\mathcal{G}_i^0$ and the child graph by $\mathcal{G}_i^1$. Note that we drop the superscripts whenever we do not need a distinction between parent and child nodes. We use $\mathcal{D} = \{(\mathcal{G}_i^0, \mathbf{s}_i^0, \mathcal{G}_i^1, \mathbf{s}_i^1) \mid i \in \{1, 2, 3, ..., N\}\}$ to denote the entire dataset.

Defining a GNN by a function $f_\theta$ that is defined over the parameter space $\Theta$, Gasse et al. (2019) proposed to find the optimal parameters by empirical risk minimization of the cross-entropy loss between the predictions and the strong branching target over the dataset $\mathcal{D}$. Thus, if there are $n$ decision variables in $\mathcal{G}_i$, $f_\theta(\mathcal{G}_i) \in \mathbb{R}^n$ represents the scores predicted by the function $f_\theta$. Denoting $\mathbf{y}_i$ as a one-hot encoded vector with the value of 1 at $y_i$ and 0 elsewhere, $\theta_y^\star$ is determined by solving

$$\theta_y^\star = \arg\min_\theta \frac{1}{N} \sum_{i=1}^N w_i \cdot CE(f_\theta(\mathcal{G}_i), \mathbf{y}_i), \tag{2}$$

where $CE$ is the cross-entropy loss function, and $w_i$ is the relative importance given to the observation $i$, which may depend on the depth of the node in the B&B tree (Gupta et al., 2020).

## 4.1 Second-best $\epsilon$-smooth loss target

Keeping the cross-entropy loss function as is, we modify the target to a smooth label $\mathbf{z}_i$, i.e., instead of one-hot encoded vector $\mathbf{y}_i$, $\mathbf{z}_i$ carries a value of $1 - \epsilon$ at the index of the strong branching target $y_i$, while the value of $\epsilon$ is equally divided among the second-best strong branching decisions in $\mathcal{Z}_i$. Thus, we obtain the optimal parameters as

$$\theta_z^\star = \arg\min_\theta \frac{1}{N} \sum_{i=1}^N w_i \cdot CE(f_\theta(\mathcal{G}_i), \mathbf{z}_i). \tag{3}$$

A modified target such as $\mathbf{z}_i$ in Eq. (3) yields parameters that tend to preserve the ranking of the second most important decisions. While intuitive and simple to implement with minimal changes in the existing framework, $\theta_z^\star$ is still not informed from the parent behavior.

## 4.2 Parent-As-Target (PAT) lookback loss term

Here, we are interested in incorporating the relation between parent and child outputs as it happens under the strong branching oracle. In doing so, we expect the learned parameters to more appropriately represent the strong branching behavior. Defining the lookback condition $L_i$ at the node $i$ as

$$L_i = \begin{cases} 1, & y_i^1 \in \mathcal{Z}_i^0 \\ 0, & \text{otherwise,} \end{cases} \tag{4}$$

we consider an additional term to Eq. (2) or Eq. (3) that enforces $f_\theta$ to follow the same ordering between parent-child nodes whenever $L_i = 1$. We call this *Parent-As-Target (PAT) lookback* term, designed to enforce similarity between the logits at the parent node for the candidates $\mathcal{C}_i^1$ of the child node, denoted as $f_\theta(\mathcal{G}_i^0)[\mathcal{C}_i^1]$, and the logits at the child node for the same candidates $\mathcal{C}_i^1$, denoted by $f_\theta(\mathcal{G}_i^1)$. Thus, we obtain $\theta_{yPAT}^\star$ with

the target $\mathbf{y}_i$ as

$$\theta^{\star}_{yPAT} = \arg\min_{\theta} \frac{1}{N} \sum_{i=1}^{N} w_i \cdot \left[ CE(f_\theta(\mathcal{G}_i^1), \mathbf{y}_i^1) + \frac{N}{\sum_{i=1}^{N} \mathbb{1}\{L_i\}} \mathbb{1}\{L_i\} \cdot \lambda_{PAT} \cdot CE(f_\theta(\mathcal{G}_i^1), f_\theta(\mathcal{G}_i^0)[\mathcal{C}_i^1]) \right], \quad (5)$$

or we obtain $\theta^{\star}_{zPAT}$ with the target $\mathbf{z}_i$ as

$$\theta^{\star}_{zPAT} = \arg\min_{\theta} \frac{1}{N} \sum_{i=1}^{N} w_i \cdot \left[ CE(f_\theta(\mathcal{G}_i^1), \mathbf{z}_i^1) + \frac{N}{\sum_{i=1}^{N} \mathbb{1}\{L_i\}} \mathbb{1}\{L_i\} \cdot \lambda_{PAT} \cdot CE(f_\theta(\mathcal{G}_i^1), f_\theta(\mathcal{G}_i^0)[\mathcal{C}_i^1]) \right]. \quad (6)$$

The first term in the brackets represents the usual cross-entropy loss which favors getting the top strong branching variable right. The second term favors aligning the predicted scores of the second-best variable in the parent node whenever it is the best in the child node $i$. This term is active only when the lookback condition is satisfied. Here, $\lambda_{PAT}$ is the relative importance given to the lookback condition, whenever it holds.

## 5 Model Selection

The dataset $\mathcal{D}$ is collected on instances of manageable size such that a reasonable number of observations are collected within a certain time budget (e.g., 1 day). We label these instances as *Small*. Given various hyperparameters involved in training machine learning models, a standard practice is to select a model with the best validation accuracy. However, owing to the sequential nature of the branching decisions, the validation accuracy is not necessarily indicative of the models' performance on other metrics such as running time. For example, due to bad branching decisions early on in the search, the models might later face sub-MILPs that are not representative of the training dataset, thereby leading to even worse decisions. Alternatively, a model might make reasonably good guesses of the strong branching variables, but the overall running time may be dominated by the solving time of the intermediate LP relaxations.

In general, a practitioner is interested in using the learned models to solve problems that are potentially bigger than those used for data collection and training. We label these instances as *Big*. To obtain some estimates of a model's ability to generalize to *Big* instances, we solve $K$ randomly generated *Medium* instances of intermediate size by using each of the learned branching strategies, $b \in \mathcal{B}$, to guide the MILP solver within a time limit of $T$ seconds per instance. The aggregate performance (e.g., arithmetic mean, geometric mean, etc.) of the branching models can be evaluated in terms of (a) solving time, denoted by $t(b)$, across $K$ instances, e.g., 1-shifted geometric mean, (b) node counts, $n(b, \mathcal{B})$, across the instances that are solved by all the strategies in $\mathcal{B}$, or (c) number of solved instances within the time limit, denoted by $w(b)$. Similarly, one can also define a metric based on the optimality gap of unsolved instances.

In exploring generalization metrics, we need to further distinguish between different objectives that a practitioner might have. For example, (a) *Minimum solving time*: the branching strategy that solves the instances as fast as possible; (b) *Maximum number of instances solved*: a branching strategy that solves the most instances to optimality within a certain time budget, irrespective of whether the strategy solved the instances fastest or not; or (c) *Minimum node count*: a branching strategy that yields the smallest trees.

Thus, even though the models are trained to mimic the strong branching oracle, thereby expecting to yield the smallest tree, we incorporate harder-to-formulate objectives by selecting the learned model using a combination of aggregate performance measures. Thus, to incorporate the objective (a), we select the branching strategy as per

$$\mathcal{B}' = \{j \mid j \in \mathcal{B}, \quad t(j) \le \min_{\mathcal{B}} t(b) + \epsilon_t\},$$
$$b^{\star}_{time} = \arg\min_{b \in \mathcal{B}'} n(b, \mathcal{B}'), \quad (7)$$

where we introduce $\epsilon_t$ as a tolerance parameter to account for the variability that is inherent to hardware-dependent measurements of solving time. For instance, $\epsilon_t = 1$ considers all the branching strategies $b \in \mathcal{B}$ with aggregate solving time within 1 second of the best such strategy, i.e., $t(b) < \min_{b \in \mathcal{B}} t(b) + 1$.

The objective corresponding to solving the largest number of instances in the minimum amount of time can be formulated as

$$
\begin{aligned}
\mathcal{B}' &= \{j \mid j \in \mathcal{B}, \quad w(j) = \max_{\mathcal{B}} w(b)\}, \\
\mathcal{B}'' &= \{j \mid j \in \mathcal{B}', \quad t(j) \le \min_{\mathcal{B}'} t(b) + \epsilon_t\}, \\
b^{\star}_{solved-time} &= \arg\min_{b \in \mathcal{B}''} n(b, \mathcal{B}''),
\end{aligned}
\tag{8}
$$

which is slightly different than the objective of selecting the strategy with the minimum time that also solves the most instances,

$$
\begin{aligned}
\mathcal{B}' &= \{j \mid j \in \mathcal{B}, \quad t(j) \le \min_{\mathcal{B}} t(b) + \epsilon\}, \\
\mathcal{B}'' &= \{j \mid j \in \mathcal{B}', \quad w(j) = \max_{\mathcal{B}'} w(b)\}, \\
b^{\star}_{time-solved} &= \arg\min_{b \in \mathcal{B}''} n(b, \mathcal{B}'').
\end{aligned}
\tag{9}
$$

We discuss other possible formulations in Appendix C.

## 6  Experiments

To evaluate the performance of our proposed methods, we consider four benchmark problem families: Combinatorial Auctions, Minimum Set Covering, Capacitated Facility Location, and Maximum Independent Set. These are the same problems that have been used extensively in the "learning to branch" literature (Gupta et al., 2020; Scavuzzo et al., 2022; Etheve et al., 2020; Sun et al., 2020) since introduced in Gasse et al. (2019). Specifically, we collect a dataset $\mathcal{D}$ for each of the problem families by solving the *Small* instances using SCIP (Gleixner et al., 2018) with the strong branching heuristic. Our models are trained to minimize the objective functions as described in Equations (2), (3), (5), or (6). Due to the space constraints, we leave the instance size, dataset collection, and training specifications to Appendices D, E, and F, respectively.

**Baselines.**  To demonstrate the utility of the proposed models, we consider three types of widely used branching strategies: (a) *Reliability Pseudocost Branching* (RPB): Given the online statistical learning aspect of this heuristic, it has been shown to be the most promising among all. The commercial solvers use this as a default branching strategy; (b) *TunedRPB*: Given that we are focused on learning a branching strategy suitable for problem sets coming from a fixed distribution, we search through the parameters of RPB to select the ones suited best for the problem family. Specifically, we run a grid search on two RPB parameters representing a trade-off between running time and the iterative performance (see Appendix G); We select the best performing parameters using the model selection framework from Section 5, making this tuned heuristic directly comparable to our method; (c) Graph Neural Networks (GNN): As proposed by Gasse et al. (2019), and widely used in the community, we use GNNs trained on the same dataset as our proposed models. These models have been shown to be the best among all the other machine learning based models. Finally, we also show the performance of our gold-standard strong branching heuristic that is used to collect the training dataset (FSB).

**Evaluation.**  We replace the variable selection heuristic in SCIP Gleixner et al. (2018) with the strategy to be evaluated. For each of the four problem families, we solve 100 randomly generated instances across three scales: *Small*, *Medium*, and *Big* (see Appendix D). Since Combinatorial Auctions' big instances are solved fairly quickly, we extend the evaluation to slightly bigger instances. Increasing scale is expected to increase the running time of the B&B algorithm. All *Small* and *Medium* instances used for evaluation are different from those used for training and model selection. Given the NP-Hard nature of the problems, we used the time limit of 45 minutes per instance to solve these instances using SCIP (Gleixner et al., 2018). See Appendix H for the specifications of SCIP and the hardware used for evaluation.

**Evaluation Metrics.** As per the standard practices in the MILP community, the performance of B&B solvers is benchmarked across the following metrics: (a) Time: 1-shifted geometric mean[1] of solving time of all the instances, irrespective of whether the instance was solved to optimality or not; (b) Nodes: 1-shifted geometric mean of the number of nodes of the *commonly solved instances* (denoted by c in parenthesis for clarity) across all branching strategies; note that this is a hardware-independent measure of performance; (c) Wins: number of instances that were solved (to optimality) the fastest by the branching strategy; (d) Solved: Total number of instances solved within the time limit; and (e) Time: 1-shifted geometric mean of solving time of the *commonly solved instances*. The commonly solved instances are a subset of instances that have been solved to optimality by all the branching strategies.

**Model Hyperparameters.** We consider a grid search over the following hyperparameters: (a) loss-target $\in \{y, z\}$, where $z$ refers to the modified loss function proposed in Section 4.1 and $y$ to the typical loss function that focuses only on the top strong branching variable; (b) $\lambda_{l_2} \in \{0.0, 0.01, 0.1, 1.0\}$, the $l_2$-regularization penalty; and (c) $\lambda_{PAT} \in \{0.01, 0.1, 0.2, 0.3\}$ to define the strength of the PAT lookback loss term proposed in Section 4.2. While we consider $\lambda_{l_2}$ for both $\theta_y$ and $\theta_z$, for $\theta_{PAT}$ we consider the best performing model among all the hyperparameters $\{$loss-target, $\lambda_{l_2}, \lambda_{PAT}\}$. For each hyperparameter configuration, we train five randomly seeded models as described in Appendix F. Finally, $\theta_y$ represents the baseline GNN from Gasse et al. (2019) without any of our proposed modifications.

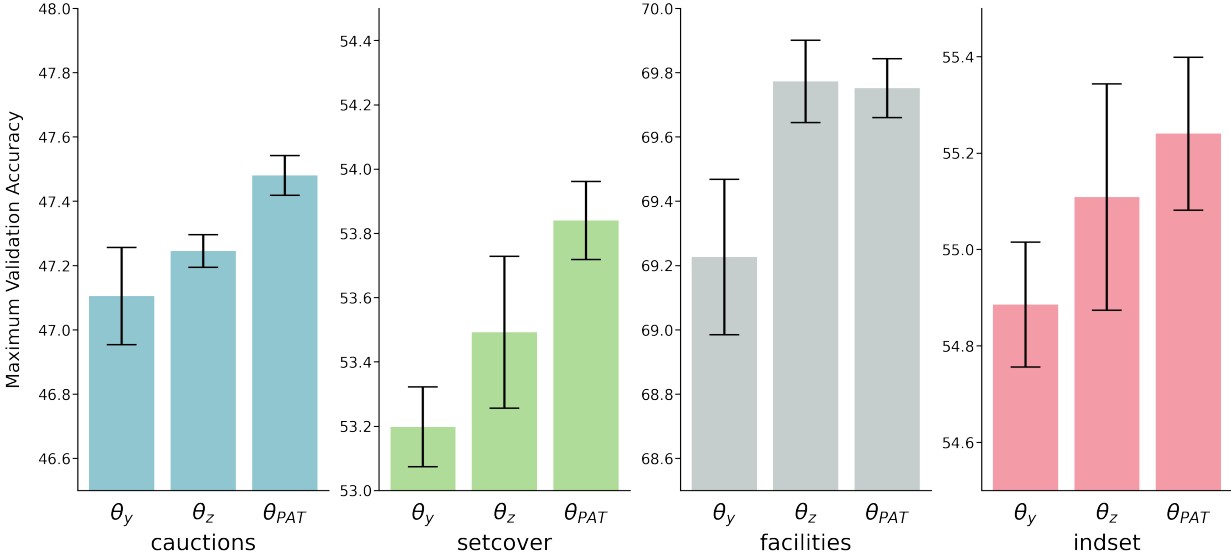

Figure 3: Maximum (across the hyperparameters) mean validation accuracy (1-standard deviation) of the proposed models is better than the baseline GNNs ($\theta_y$). We see that the models trained with smoothed target ($\theta_z$) and those with PAT lookback loss term ($\theta_{PAT}$) result in better validation accuracy.

## 6.1 Results

**Hyperparameter Selection.** Figure 3 shows validation accuracy of the best performing hyperparameters for $\theta_y$, $\theta_z$, and $\theta_{PAT}$ according to the validation accuracy on collected dataset $\mathcal{D}$. However, to select the models based on the generalization performance, we solve 100 randomly generated medium instances and gather the metrics as described above. Table 1 shows the selection of hyperparameters for $\theta_{PAT}$ as per various criteria (see Appendix I for the best parameters for $\theta_y$ and $\theta_z$). We observe that the models selected by validation accuracy are not always preferred by the selection criteria defined on the evaluation of medium instances. Second, we observe that Eq. (8) and (9) may or may not have consensus among them; a strategy might have the fastest solving times for all instances except one, but other strategies might solve all the instances in just slightly more time. Third, we observe that the time limit per instance, $T$, does play a role in model selection;

---

[1]For a complete definition, refer to Appendix A.3 in Achterberg (2007)

both facilities and indset have a different preference. Finally, we observe that even though the modified target $z$ is not preferred by all the problem families, there is a consensus for the use of the PAT lookback loss term.

Table 1: Best performing hyperparameters {loss-target, $\lambda_{l_2}, \lambda_{PAT}$} for $\theta_{PAT}$. We see that loss-target $= z$ is preferred by some problem families, while the PAT lookback term is preferred by all. In addition, the model selection criteria does impact the chosen hyperparameters.

| Problem family | Validation Accuracy | Eq. (8)($T = 30$mins) | Eq. (9)($T = 30$mins) | Eq. (8)($T = 15$mins) |
|---|---|---|---|---|
| cauctions | $\{y, 0.0, 0.01\}$ | $\{z, 0.0, 0.1\}$ | $\{z, 0.0, 0.1\}$ | $\{z, 0.0, 0.1\}$ |
| setcover | $\{y, 0.0, 0.1\}$ | $\{y, 0.0, 0.1\}$ | $\{y, 0.0, 0.1\}$ | $\{y, 0.0, 0.1\}$ |
| facilities | $\{z, 0.0, 0.0\}$ | $\{z, 0.0, 0.1\}$ | $\{z, 0.0, 0.1\}$ | $\{z, 0.0, 0.3\}$ |
| indset | $\{y, 0.0, 0.1\}$ | $\{z, 0.01, 0.2\}$ | $\{y, 0.1, 0.3\}$ | $\{y, 0.1, 0.3\}$ |

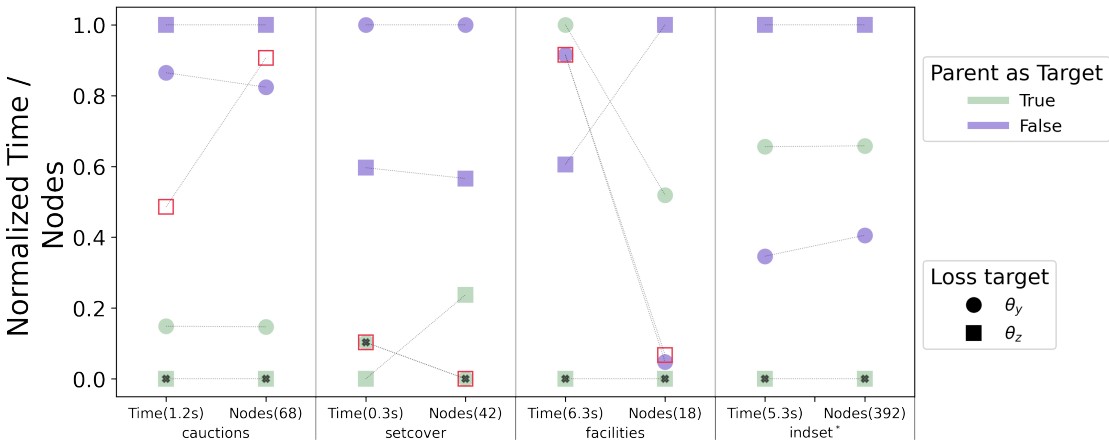

Figure 4: We plot the range-normalized (range is specified in parenthesis) Time and Node performance of the selected models as per Eq. (8). The centered "X" black mark shows the final models that were selected to be used for evaluating the performance on bigger instances. The points with a red outline show the performance of the models selected according to the best validation accuracy (Note that we omit such models for indset as it distorts the scale of the plot; see Appendix J for complete data).

**Validation performance on Medium instances.** To assess the interplay between our proposals in Section 4 and the model selection framework proposed in Section 5, we compare the performance of the selected models using Eq. (8) ($T = 30$ minutes) for each of $\theta_y$, $\theta_z$, $\theta_{yPAT}$, and $\theta_{zPAT}$. Figure 4 compares their performance on medium instances with respect to Time and Nodes. To accommodate different scales of time and nodes, we plot the range-normalized values of these measures and show the range of these measures in parentheses. The centered black marks show the final models selected as per Eq. (8). The models chosen as per validation accuracy are shown with transparent marks with red border.

We make the following observations. First, the PAT lookback term is beneficial for the generalization performance most of the time. We observe that for the facilities set of problems, the current GNNs already respect the lookback condition sufficiently well that the proposed modifications do not yield significant improvements compared to $\theta_y$. Second, we see that the cauctions and setcover models perform equally well with respect to time, thereby making the node count an important criterion for identifying better branching models. This is especially important because time measurements are hardware-dependent and thus not as reliable. Third, central to the motivation of our model selection framework, the models chosen as per validation accuracy do not fare well on practical objectives such as Time and Nodes.

**Evaluation on Big instances.** Given that sets of *Small* and *Medium* instances have already been used in training and selection of the final models, we leave the evaluation on additional sets of unseen instances from these families to Appendix L. Here, we evaluate the performance of the selected models (as per Eq. (8)($T = 30$

minutes)) on bigger instances. We use the same instance scaling scheme as proposed by Gasse et al. (2019) (see Appendix D). Table 2 shows various evaluation metrics as computed from the evaluation of 100 randomly generated *Big* instances. Since *Big* instances of Combinatorial Auctions are solvable by all the strategies, we extend the scale of these instances to *Bigger* instances. Specifically, we observe that $\theta_{PAT}$ (GNN-PAT) outperforms the baseline model (GNN) in all the problem families on all fronts – Time, Wins, Solved, and Nodes. As an example, for Maximum Independent Set problems, we observe a 15% decrease in Time and a 22% decrease in Nodes. Notably, GNN-PAT increases the number of "Solved" instances by 4 to 5 for all but one problem family. Solving additional instances to optimality is a testament to the improved branching decisions brought about by GNN-PAT.

Finally, as noticed above, the learned models for Maximum Independent Set might result in a different hyperparameter configuration based on the selection criterion. Therefore, we compare the performance of the GNN-PAT models that are selected by each of the Eqs. (8) and (9) against GNN in Table 3. We observe that as per the selection criterion of Eq. (8), the branching strategy solved the most number of instances. However, as Eq. (9) prefers the strategy with overall lower running time, we observe superior performance of the branching strategy on Time of commonly solved instances. These observations confirm that the proposed model selection approach yields the expected outcomes on unseen test instances.

Similarly, we look at the effect of the time limit $T$ on the selection criterion. The models selected as per Eq. (8) for facilities differ depending on the specified time limit $T$ (see Table 1). We look at the how the

Table 2: Performance of *branching strategies* on *Big* evaluation instances. The best performing numbers are in **bold**. Refer to Section 6 for metrics. Since all of the *Big* Combinatorial Auctions instances were solved to optimality, we extend the evaluation to *Bigger* instances (*since only 10 instances were solved using FSB, we omit it here; See Appendix K for the full results including those on *Big* instances.)

| Model | Time | Time (c) | Wins | Solved | Nodes (c) |
|---|---|---|---|---|---|
| FSB* | n/a | n/a | n/a | n/a | n/a |
| RPB | 626.81 | 434.92 | 1 | 80 | 17 979 |
| TUNEDRPB | 644.20 | 450.06 | 0 | 80 | 18 104 |
| GNN | 507.06 | 333.59 | 14 | 80 | 17 145 |
| GNN-PAT (ours) | **477.26** | **310.22** | **69** | **84** | **16 388** |
| Combinatorial Auction (Bigger) | | | | | |
| FSB | 2700 | n/a | 0 | 0 | n/a |
| RPB | 1883.32 | 1213.57 | 1 | 47 | 58 766 |
| TUNEDRPB | 1851.83 | 1168.58 | 0 | 48 | 58 155 |
| GNN | 1708.99 | 991.07 | 5 | 54 | 39 535 |
| GNN-PAT (ours) | **1601.28** | **892.85** | **54** | **59** | **38 385** |
| Set Covering | | | | | |
| FSB | 917.39 | 758.19 | 8 | 85 | 50 |
| RPB | 737.66 | 607.19 | 3 | 92 | 104 |
| TUNEDRPB | 751.27 | 619.27 | 2 | 91 | **97** |
| GNN | 646.03 | 525.80 | 16 | 92 | 293 |
| GNN-PAT (ours) | **581.91** | **471.20** | **66** | **95** | 304 |
| Capacitated Facility Location | | | | | |
| FSB | 2700 | n/a | 0 | 0 | n/a |
| RPB | 1984.91 | 888.04 | 0 | 32 | 12 407 |
| TUNEDRPB | 2016.85 | 952.25 | 0 | 33 | 12 940 |
| GNN | 1207.62 | 279.71 | 14 | 65 | 7934 |
| GNN-PAT (ours) | **1035.32** | **233.97** | **56** | **70** | **6122** |
| Maximum Independent Set | | | | | |

time limit might affect generalization performance in Table 3. Specifically, we observe that insufficient time to evaluate *Medium* instances may lead to suboptimal hyperparameters. To conclude, we've shown that the model selection criterion will impact generalization performance significantly.

## 7   Discussion

An objective in this paper was to imitate the strong branching behavior more closely by taking advantage of its lookback property. The proposals in Section 4 are aimed at doing so. A post-hoc analysis shows up to 16% improvement in the GNNs' ability to follow the lookback property (see Appendix M). Such improvements are evident at various depths from the root node (see Appendix N).

We can argue that the proposals in Section 4 are regularizers or inductive biases. Although the modified target and the PAT lookback term were inspired to induce the required oracle behavior, owing to the lack of interpretability of GNNs, we cannot attribute the GNN-PATs' performance improvements to the ability to follow the lookback property with 100% certainty. However, if we consider that the proposals did cause the

observed improvements, we might consider it as an inductive bias. Inductive biases are defined as any pre-coded knowledge about the target behavior, such as neural network architecture, choice of features, or loss function.

Further, the PAT lookback loss term is similar to the objective function in knowledge distillation (Hinton et al., 2015), i.e., we are minimizing the cross-entropy between the logits of one model (child node) and the other model (parent node). Whereas in the original distillation framework, there are two separate models (teacher and student) that act on the same inputs, the PAT lookback term

Table 3: Performance measures on branching strategies selected as per different criteria specified in Eqs. (8) and (9) and the specified time limit per instance $T$. We observe that each criterion supports the respective measure on scaled-up instances.

| Model | Time | Time (c) | Wins | Solved | Nodes |
|---|---|---|---|---|---|
| GNN | 1207.62 | 753.81 | 1 | 65 | 29 573 |
| GNN-PAT (Eq. 8) | **1035.32** | 621.42 | **38** | **70** | **23 574** |
| GNN-PAT (Eq. 9) | 1063.12 | **613.96** | 31 | 66 | **21 162** |
| Maximum Independent Set | | | | | |
| GNN | 646.03 | 562.20 | 13 | 92 | 314 |
| GNN-PAT (Eq. (8)(T=30mins)) | **581.91** | **503.85** | **58** | **95** | **326** |
| GNN-PAT (Eq. (8)(T=15mins)) | 635.04 | 551.53 | 24 | 94 | 388 |
| Capacitated Facility Location | | | | | |

can be understood as a form of *cross-distillation* that acts through the same model but on different observations which share some characteristics (e.g., output logits). Thus, we can understand that the PAT lookback term distills knowledge from the parent node to the child node, as was intended.

Finally, our model selection framework enables us to incorporate complex metrics like Time and Nodes into hyperparameter selection criterion. We hope that this framework will help aligning the research in machine learning methods for MILP solvers with practitioners' varied objectives.

**Limitations.** As illustrated in Figure 7, the GNNs for facilities instances are capable of capturing the lookback condition 80% of the time. Since it is not possible to consider all possible problem families and their varied formulations, we cannot make a definitive claim on whether our proposed modifications will be useful all the time. Therefore, we recommend checking for the prevalence of the lookback condition to get some idea of expected improvement.

Further, due to time and resource constraints, we restricted the evaluation for model selection to just 100 *Medium* instances. However, for a more robust selection, we suggest using a larger set of instances. One can also consider setting the size of *Medium* instances such that majority of them are solved, thereby resulting in robust selection of better performing hyperparameters; see Table 1 on how the best hyperparameters vary according to time limit per instance $T$ and Table 3 for the effect of $T$ on generalization performance.

Finally, we emphasize that the model selection criteria are very much dependent on how these models will be deployed. For example, a practitioner might only be concerned with solving the maximum number of instances (to optimality) while ensuring the smallest optimality gaps in the unsolved instances. This objective can be formulated within our proposed framework, but considering all such formulations is beyond the scope of this work.

**Future work.** Although we did not specify the minimum optimality gap as the objective of the branching strategies, we ran a post-hoc analysis to compare 1-shifted geometric mean of optimality gaps of the commonly unsolved instances (lower is better). Figure 5 shows that, except for setcover instances, the proposed branching strategies are able to close larger gaps than the rest. We acknowledge that, depending on the use, the optimality gap might be of primary importance to the practitioner. We think that the exploration of optimality gap as a secondary objective could be a subject of future work.

As evident from the gaps in Figure 7 (Appendix M) and Figure 8 (Appendix N), we plan to design more ways to incorporate the lookback condition explored in this work. While we studied how the parent and child nodes in a B&B tree are related with respect to a simple PAT lookback condition, there still exists several ways in which such nodes can be related (see Appendix O for another example). Thus, the design of machine learning algorithms to *discover* and *exploit* such dependencies could be an important direction for future research. Moreover, such machine learning-aided discoveries can be equally important for the MILP

community to inspire the design of novel heuristics or improve the existing manually-designed heuristics applicable to general instances.

## 8 Conclusion

With the huge gap between the performance of deep learning based heuristics and the oracle heuristics, we expect that the research efforts might require more in-depth investigation of how to imbue these models with the same "reasoning" as the oracles themselves. In this line of thought, we investigated how the parent-child nodes of a B&B tree are related to each other under the oracle heuristic. We found that quite often, the parent's second-best choice is the child's best choice. To incorporate this lookback condition into model training, we designed two methods to align the models more closely with the strong branching oracle's behavior. We believe that this investigative approach to imitating oracle behavior could be a useful way to close the gap between machine learning and the oracle heuristics.

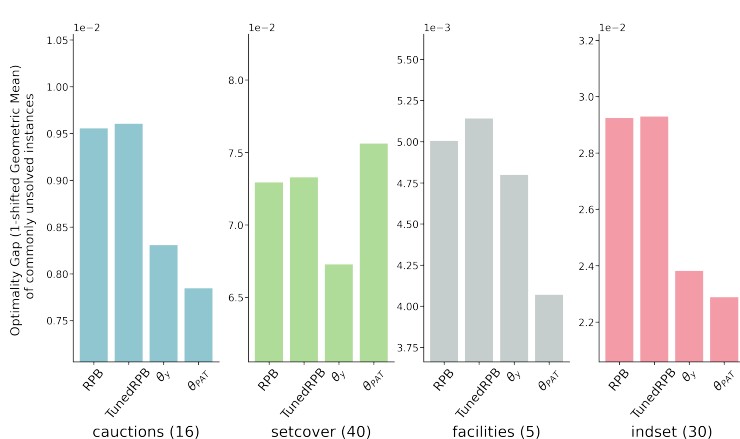

Figure 5: Post-hoc analysis of optimality gap of commonly unsolved instances (number is shown in parentheses next to the problem family label) shows that $\theta_{PAT}$ is able to achieve the best optimality gap (except for setcover) even though it is not the primary objective specified in the model training.

### Broader Impact Statement

This paper continues the exploration of the use of machine learning techniques for the most critical step in branch-and-bound methods, i.e., variable selection. Branch and bound is the method of choice for solving a myriad of discrete optimization problems appearing in all sort of applications (a few named in the introduction of this paper) and it is the basic scheme of all commercial and open-source discrete optimization solvers. Thus, the impact of the research in the area is potentially very high, not only from a methodological perspective but also in terms of day to day challenges that we all face, including drug discoveries and climate change.

This paper significantly advances the research in the area by observing for the first time a hidden pattern, the lookback phenomenon, in the statistical evolution of the most successful heuristic for variable selection. The paper proposes several methods to exploit such a phenomenon and makes a significant step forward on the use of ML for discrete optimization.

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

## Appendix

## A    Lookback property on some real-world instances

We look at the frequency of the lookback property on some of the real-world instances. Specifically, we study this property in the context of MIP instances related to wildlife management, first proposed by Dilkina et al. (2017). These instances have been used by the MIP community to demonstrate the efficacy of various proposals on the real-world MIP instances Nair et al. (2020); Hutter et al. (2010); He et al. (2014a). Table 4 shows that at least 30% of the parent-child pairs exhibit this property.

Table 4: Frequency of the lookback property in the real-world instances is as prevalent as in the synthetic instances considered in the main paper. These instances are made available by Dilkina et al. (2017).

| Instances | Description | number of parent-child pairs collected | number of parent-child pairs exhibiting the lookback property | Frequency of the lookback property |
|---|---|---|---|---|
| CORLAT | Corridor planning in wildlife management | 5082 | 1765 | 34.73% |
| RCW | Red-cockaded woodpecker diffusion conservation | 5115 | 1952 | 38.16% |

## B    Lookback property as a function of depth-decile

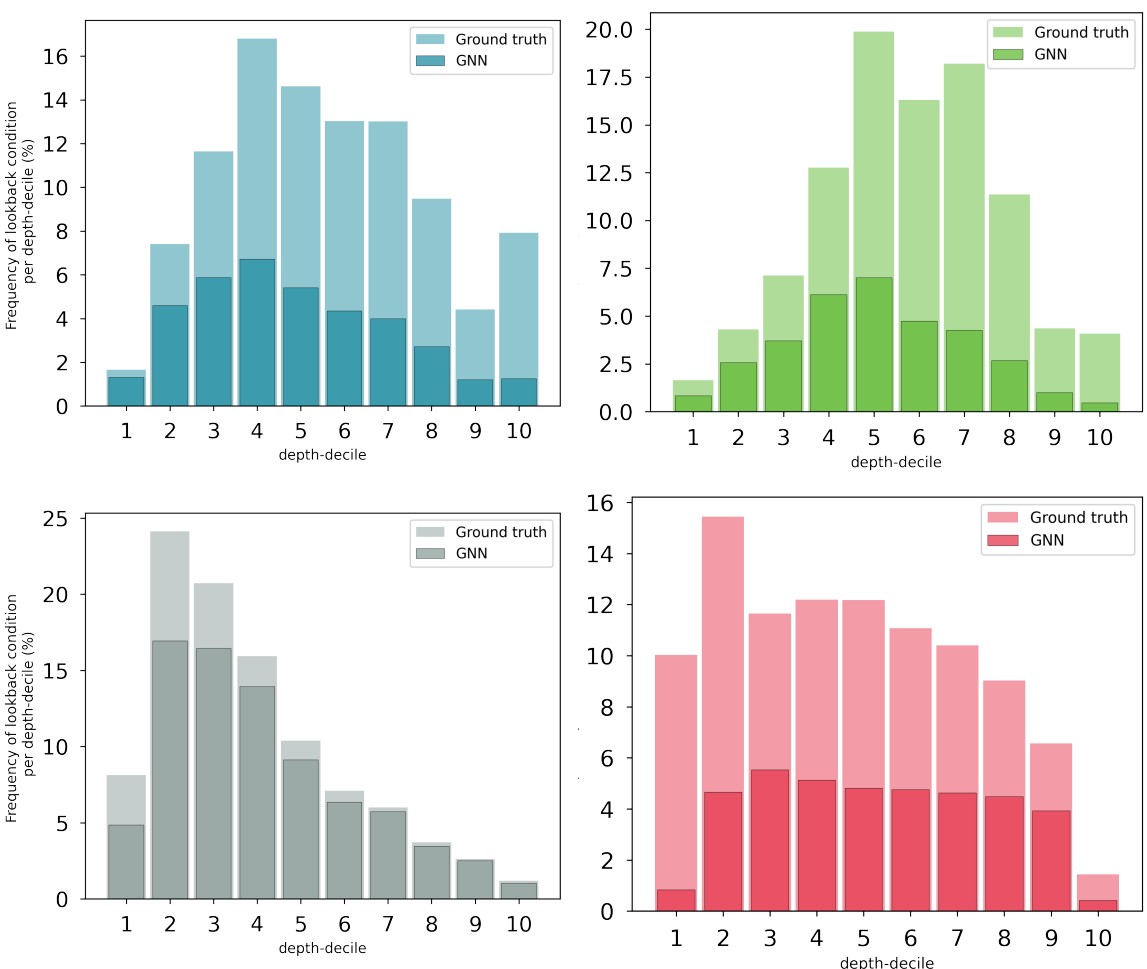

Figure 6: The gap between the frequency of the lookback condition per *depth-decile* for the strong branching oracle (Ground truth) and the traditionally trained GNNs (GNN) presents an opportunity to improve the GNN models. See Section 4 for the dataset collection procedure.

## C   Model Selection Objectives

To incorporate the objective of maximum solved instances, we select the branching strategy as per,

$$
\mathcal{B}^{'} = \{j \mid j \in \mathcal{B}, \quad w(j) = \max_{\mathcal{B}} w(b)\},
$$
$$
b^{\star} = \underset{b \in \mathcal{B}'}{\arg\min}\, n(b, \mathcal{B}^{'})
\tag{10}
$$

Finally, to select the strategy only with the minimum node count, we select the branching strategy as per,

$$
b^{\star} = \underset{b \in \mathcal{B}}{\arg\min}\, n(b, \mathcal{B})
\tag{11}
$$

## D   Instance Specifications

We follow the procedure outlined by Gasse et al. (2019) to generate and scale the instances. A well commented and functional code to generate these instances can be found on the authors' Github repository[2]. For convenience, we describe the procedure here.

---

[2]https://github.com/ds4dm/learn2branch/blob/master/01_generate_instances.py

### D.1 Combinatorial Auction

These instances are generated following the arbitrary scheme described in the section 4.3 of Leyton-Brown et al. (2000). The scalable parameters are the number of items and the number of bids.

Table 5: Parameters for Combinatorial Auctions

| Instance Size | number of items | number of bids | use |
|---|---|---|---|
| Small-Training | 100 | 500 | Dataset Collection & Training |
| Small-Validation | 100 | 500 | Dataset Collection & Validation |
| Medium-Validation | 200 | 1000 | Validation Evaluation |
| Small | 100 | 500 | Test Evaluation |
| Medium | 200 | 1000 | Test Evaluation |
| Big | 300 | 1500 | Test Evaluation |
| Bigger | 350 | 1750 | Test Evaluation |

### D.2 Set Covering

These instances are generated using the method described in Balas & Ho (1980). The scalable parameters are number of items, where the number of sets are fixed to 1000.

Table 6: Parameters for Minimum Weighted Set Cover. Number of sets is fixed to 1000.

| Instance Size | number of items | use |
|---|---|---|
| Small-Training | 500 | Dataset Collection & Training |
| Small-Validation | 500 | Dataset Collection & Validation |
| Medium-Validation | 1000 | Validation Evaluation |
| Small | 500 | Test Evaluation |
| Medium | 1000 | Test Evaluation |
| Big | 2000 | Test Evaluation |

### D.3 Capacitated Facility Location

These instances are generated the method described by Cornuéjols et al. (1991). Fixing the number of facilities to 100, the scalable parameter is the number of customers.

Table 7: Parameters for Capacitated Facility Location. Number of facilities is fixed to 100.

| Instance Size | number of customers | use |
|---|---|---|
| Small-Training | 100 | Dataset Collection & Training |
| Small-Validation | 100 | Dataset Collection & Validation |
| Medium-Validation | 200 | Validation Evaluation |
| Small | 100 | Test Evaluation |
| Medium | 200 | Test Evaluation |
| Big | 400 | Test Evaluation |

### D.4 Maximum Independent Set

These instances are generated by formulating the maximum independent set problem on a randomly generated Barabási-Albert with edge probability of 0.4. The scalable parameter is the number of nodes.

Table 8: Parameters for Maximum Independent Set. Affinity is fixed to 4.

| Instance Size | number of nodes | use |
|---|---|---|
| Small-Training | 750 | Dataset Collection & Training |
| Small-Validation | 750 | Dataset Collection & Validation |
| Medium-Validation | 1000 | Validation Evaluation |
| Small | 750 | Test Evaluation |
| Medium | 1000 | Test Evaluation |
| Big | 1500 | Test Evaluation |

## E  Data Generation

For each of the problem family, we generate 10,000 *Small* random instances to collect training data, 2,000 *Small* random instances to collect the validation data, and 20 *Medium* instances for model selection. We use SCIP Gleixner et al. (2018) with strong branching heuristic to solve the randomly generated instances and collect data of the form $\mathcal{D} = \{(\mathcal{G}_i^0, \mathbf{s}_i^0, \mathcal{G}_i^1, \mathbf{s}_i^1) \mid i \in \{1, 2, 3, ..., N\}\}$, as described in the main text. We collected a total of 150,000 training observations and 30,000 validation observations.

The hand-engineered features to the GNNs are same as Gasse et al. (2019). For a full description of these features, pleasee see Section 2 in Supplementary material of Gupta et al. (2020).

## F  Training Specifications

Our models are all implemented in PyTorch (Paszke et al., 2017). Following Gupta et al. (2020), we didn't change any of the training parameters, for example, we used the Adam (Kingma & Ba, 2014) optimizer with the learning rate of $1e^{-3}$, training batch size of 32, and a learning rate scheduler to reduce the learning rate by a factor of 0.2 in the absence of any improvement in the validation loss for the last 15 epochs Moreover, we use the early stopping criterion to stop the training if the validation loss doesn't improve over 30 epochs. We validate the performance of model on the validation dataset after every epoch consisting of 10K random training samples. For each problem family, we trained models with five random seeds.

## G  TunedRPB

We consider two parameters in RPB to negotiate the trade-off between the compactness of the resulting B&B tree and the computational time. Broadly, RPB performs strong branching on MaxLookahead candidates until it has collected enough information to reliably act on it. The selection of candidates is prioritized by the reliability of *pseudoscores* (statistically learned score to estimate bound improvement per fractional rounding of the variable) collected during the strong branching operations. If the minimum psuedoscore obtained by withere rounding up or rounding down of the integer-constrained vairable is less than MaxReliable, the candidate is deemed unreliable, thereby prioritizing it in the next rows. Thus, we observe the following trade-off by varying these two parameters: (i) MaxReliable: lower values prefer faster solving times at the expense of larger trees, and (ii) MaxLookahead: higher values prefer shorter trees at the expense of more computational time.

To have a version of RPB that is trained on the training instances of interest, we run a hyperparameter grid-search on the following values -

1. MaxLookahead: $\{6, 7, 8, 9, 10, 11\}$

2. MaxReliable: $\{3, 4, 5, 6, 7, 8\}$

Specifically, we followed the procedure as described in Section 5 to solve 100 randomly generated medium instances, and select the best performing hyperparameters according to Eq. (8).

## H   Evaluation Specification

The evaluation on *Big* instances is performed by using SCIP 6.0.1 installed on an Intel(R) Xeon(R) CPU E5-2650 v4 @ 2.20GHz. The learned neural networks, as branching models, are run on NVIDIA-TITAN Xp GPU card with CUDA 10.1. All of the main evaluations are done, as is a standard practice, while ensuring that the ratio of the number of cores on the machine to the load average is more than 4. This condition ensures that each process on the machine has at least 4 CPUs at a time.

The model selection is carried out by solving *Medium* instances on the shared cluster as specified by Nair et al. (2020) (see Section 12.7). Specifically, we solve a benchmark MIPLIB problem ('vpm2.mps') every 60 seconds to collect the solving time statistics. Given the solving times of the benchmark problem on the reference machine, we recalibrate the solving time of the instance, which is used as the final running time. We solve 20 independently generated *Medium* instances with five seeds resulting in 100 independent evaluations.

We followed the standard procedure used proposed by Gasse et al. (2019) to set SCIP for evaluating branching strategies. Specifically, we used the following SCIP parameters to evaluate the branching strategies

1. Cutting planes are allowed only at the root node

2. No restarts are allowed

## I   Hyperparameters & Model Selection

We use the ridge regression to penalize the parameters $\theta_y$ and $\theta_z$. In addition to $\lambda_{l2}$, $\theta_{PAT}$ searches over *target* and $\lambda_{PAT}$. The following values were used for the hyperparameter search -

1. $\lambda_{l2} = \{0.01, 0.1, 1.0\}$

2. $\lambda_{PAT} = \{0.01, 0.1, 0.2, 0.3\}$

3. $target = \{\mathbf{y}, \mathbf{z}\}$

Finally, Table 9 shows the best performing hyperparameters according to Eq. (8) for $\theta_y$, $\theta_z$, and $\theta_{PAT}$. As observed in Gupta et al. (2020), we found that the $l2$-regularization is useful for the performance of indset models.

Table 9: Best performing hyperparameters according to Eq. (8)

| problem family | $\theta_y\{\lambda_{l2}\}$ | $\theta_z\{\lambda_{l2}\}$ | $\theta_{PAT}\{target, \lambda_{l2}, \lambda_{PAT}\}$ |
|---|---|---|---|
| cauctions | 0.0 | 0.0 | $\{z, 0.0, 0.1\}$ |
| setcover | 0.0 | 0.0 | $\{y, 0.0, 0.1\}$ |
| facilities | 0.0 | 0.0 | $\{z, 0.0, 0.1\}$ |
| indset | 0.1 | 0.1 | $\{z, 0.01, 0.2\}$ |

## J   Performance on medium validation instances

Table 10 shows the data that was used to plot the points in Figure 4. Due to the huge variability in the performance metrics of indset we omit the performance of $\theta_{accuracy}$ from Figure 4.

Table 10: Data for Figure 4

| Model | Metric | cauctions | setcover | facilities | indset |
|---|---|---|---|---|---|
| $\theta_y$ | Time | 14.66 | 38.21 | 159.60 | 73.16 |
|  | Nodes | 1077 | 1546 | 421 | 3102 |
| $\theta_z$ | Time | 14.82 | 38.08 | 157.64 | 76.61 |
|  | Nodes | 1089 | 1527 | 437 | 3335 |
| $\theta_{yPAT}$ | Time | 13.83 | 37.92 | 160.13 | 74.80 |
|  | Nodes | 1031 | 1503 | 429 | 3201 |
| $\theta_{zPAT}$ | Time | 13.65 | 37.89 | 153.81 | 71.32 |
|  | Nodes | 1022 | 1513 | 420 | 2943 |
| $\theta_{accuracy}$ | Time | 14.22 | 37.92 | 159.59 | 572.39 |
|  | Nodes | 1083 | 1503 | 421 | 9430 |

## K  Full performance of Combinatorial Auction models on Big and Bigger instances

Since *Big* Combinatorial Auction instances are solved by all the strategies, we show them in Table 11. We extend the size of these instances to *Bigger* and show the evaluation in the main paper without FSB as there are only 5 instances solved distorting the comparison of Time (c) and Node (c). See Table 12 for the full results.

Table 11: Performance of variable selection strategies on *Big* instances for Combiatorial Auctions instances. See Appendix D for the instance scaling parameters.

| Model | Time | Time (c) | Wins | Solved | Nodes (c) |
|---|---|---|---|---|---|
| FSB | 2075.60 | 1643.78 | 0 | 53 | 336 |
| RPB | 202.78 | 113.47 | 1 | 100 | 4640 |
| TUNEDRPB | 205.57 | 112.80 | 0 | 100 | 4611 |
| GNN | **139.30** | **72.17** | **64** | 100 | 4142 |
| GNN-PAT (ours) | 141.63 | 73.31 | 35 | 100 | **3754** |

Table 12: Performance of variable selection strategies on *Bigger* instances for Combinatorial Auctions instances. See Appendix D for the instance scaling parameters.

| Model | Time | Time (c) | Wins | Solved | Nodes (c) | Optimality Gap (16) |
|---|---|---|---|---|---|---|
| FSB | 2591.81 | 1793.64 | 0 | 10 | 257 | 0.033 102 |
| RPB | 626.81 | 88.22 | 1 | 80 | 5719 | 0.009 555 |
| TUNEDRPB | 644.20 | 97.72 | 0 | 80 | 6191 | 0.009 605 |
| GNN | 507.06 | **59.22** | 14 | 80 | **5630** | 0.008 307 |
| GNN-PAT (ours) | **477.26** | 65.62 | **69** | **84** | 5757 | **0.007 844** |

## L  Performance on small and medium instances

Table 13 shows the performance of the selected strategies as per Eq. (8) on small and medium instances. This table is a counterpart to the Table 2.

Table 13: Performance of *branching strategies* on evaluation instances. We report geometric mean of solving times, number of times a method won (in solving time) over total finished runs, and geometric mean of number of nodes. Refer to section 6 for more details. The best performing results are in **bold**.

| Model | Small | | | | Medium | | | |
|---|---|---|---|---|---|---|---|---|
| | Time | Wins | Solved | Nodes | Time | Wins | Solved | Nodes |
| FSB | 5.85 | 0 | 100 | 6 | 127.56 | 0 | 100 | 72 |
| RPB | 3.89 | 0 | 100 | **11** | 25.31 | 0 | 100 | 696 |
| TUNEDRPB | 3.72 | 0 | 100 | **11** | 24.70 | 0 | 100 | **591** |
| GNN | **2.10** | **82** | 100 | 71 | **13.15** | 58 | 100 | 693 |
| GNN-PAT (ours) | 2.18 | 18 | 100 | 72 | 13.25 | 42 | 100 | 654 |
| | | | | Combinatorial Auction | | | | |
| FSB | 26.17 | 0 | 100 | 17 | 531.47 | 0 | 75 | 117 |
| RPB | 13.41 | 0 | 100 | 54 | 91.33 | 0 | 100 | 1119 |
| TUNEDRPB | 13.63 | 0 | 100 | **48** | 93.60 | 0 | 100 | 1131 |
| GNN | 9.37 | 5 | 100 | 136 | 64.81 | 1 | 100 | 1030 |
| GNN-PAT (ours) | **9.03** | **95** | 100 | 134 | **58.48** | **99** | 100 | **997** |
| | | | | Set Covering | | | | |
| FSB | 41.61 | 3 | 100 | 14 | 264.67 | 3 | 98 | 73 |
| RPB | 36.58 | 3 | 100 | 22 | 206.28 | 1 | 100 | 147 |
| TUNEDRPB | 37.56 | 1 | 100 | **21** | 211.77 | 2 | 100 | **140** |
| GNN | **27.20** | **78** | 100 | 113 | **146.41** | **64** | 100 | 320 |
| GNN-PAT (ours) | 29.46 | 15 | 100 | 112 | 159.95 | 30 | 100 | 329 |
| | | | | Capacitated Facility Location | | | | |
| FSB | 626.33 | 0 | 93 | 54 | 1634.40 | 0 | 60 | 46 |
| RPB | 58.03 | 0 | 100 | 702 | 144.92 | 0 | 100 | 770 |
| TUNEDRPB | 56.99 | 1 | 100 | 697 | 143.88 | 2 | 100 | **748** |
| GNN | 35.04 | 28 | 100 | 1000 | 76.53 | 36 | 100 | 795 |
| GNN-PAT (ours) | **31.96** | **71** | 100 | **455** | **72.00** | **62** | 100 | 789 |
| | | | | Maximum Independent Set | | | | |

## M  Post-hoc analysis of the lookback property

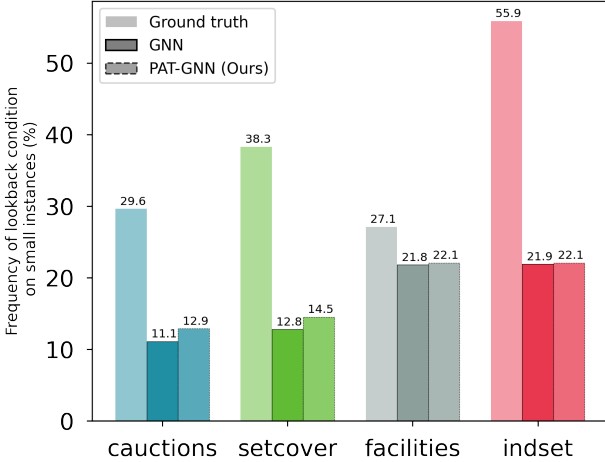

Figure 7: Statistics of the lookback condition as observed when the problems are solved using the strong branching oracle (Ground truth). We also show how many times traditional GNNs (GNN) and our proposed GNNs (PAT-GNN) respect the lookback condition on the same dataset. Note that the displayed statistics are on the offline dataset and do not reflect the final inference-time performance of the models. We find that the small improvements shown here results in large gains in the inference time performance (see Section 6)

## N  Post-hoc depth-decile analysis of the lookback property

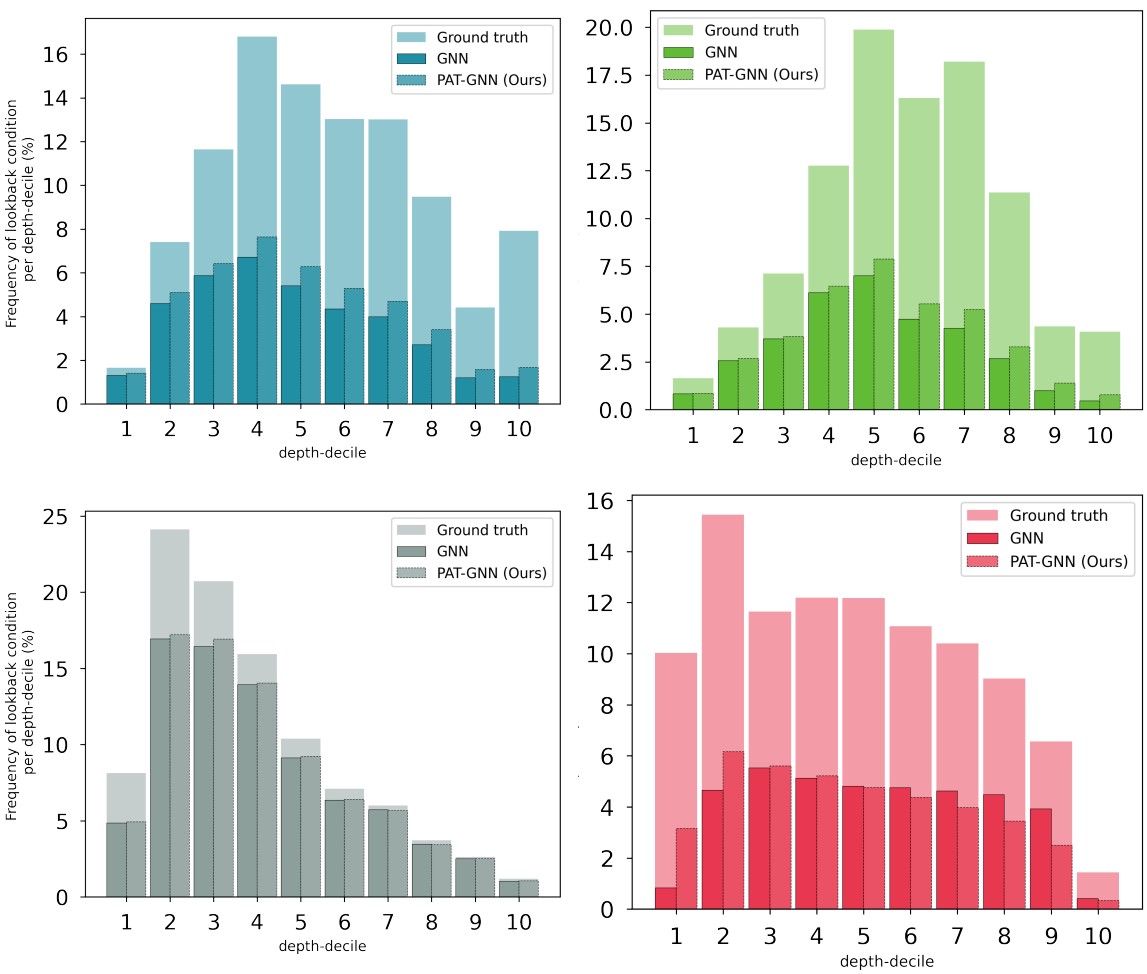

Figure 8: The gap between the frequency of the lookback condition per *depth-decile* for the strong branching oracle (Ground truth) and the traditionally trained GNNs (GNN) presents an opportunity to improve the GNN models (PAT-GNN). See Section 4 for the dataset collection procedure.

## O   Same strong branching decisions at the sibling nodes

Any MILP solving procedure results in a tree of sub-problems, where each node shares some characteristics with its parent. This line of resemblance can eventually be traced all the way back to the original MILP. For example, the bipartite graph structure remains the same throughout the tree, a fact exploited by Gupta et al. (2020) to design CPU-efficient GNN-based models for learning to branch.

We investigate more of such dependencies among the B&B tree nodes of problem instances from the benchmark problem families (Gasse et al., 2019) (see Appendix D for more details), which we label as cauctions (Combinatorial Auctions), setcover (Minimum Set Cover), facilities (Capacitated Facility Location), and indset (Maximum Independent Set).

Specifically, we investigate the frequency with which sibling nodes in the B&B tree have the same strong branching decision. Figure 9 shows that this condition happens between 3-7% of the times on the small instances that were used to collect approximately 30K observations. The state-of-the-art GNNs (Gasse et al., 2019) are not able to capture this dependency as well. However, given that this condition is fairly less frequent relative to the *lookback* condition, we do not explore this condition in our current work.

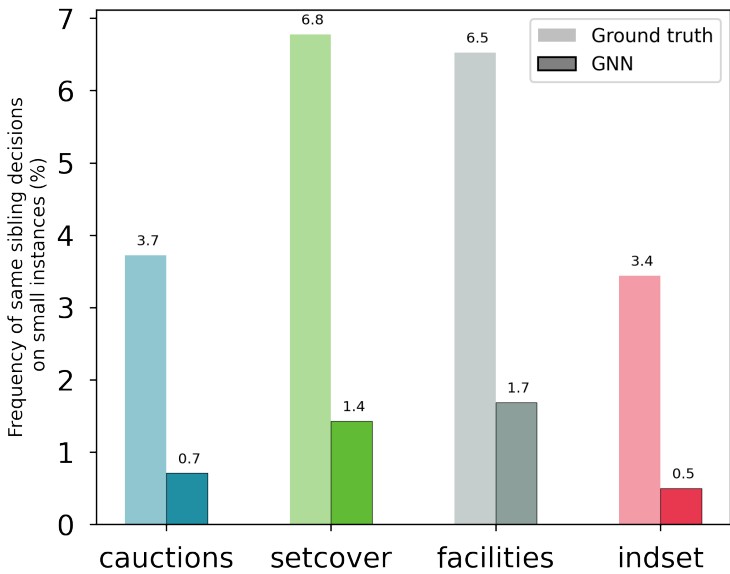

Figure 9: Frequency with which sibling nodes in a B&B tree have the same strong branching decisions (Ground truth), and the fraction of times this condition is satisfied by GNNs trained as per Gasse et al. (2019).

## P Performance comparison with default SCIP

In this section, we provide comparison of our branching strategies with the default SCIP and the tuned version of SCIP following the benchmarking procedure of Nair et al. (2020). But it is important to note that SCIP comes loaded with several heuristics, e.g., cut selection, primal heuristics, restarts, etc. To study the effect of branching strategies, the standard practice in the community is to switch off these heuristics. The results in the main paper are based on this choice. Thus, comparing the branching strategies with default SCIP isn't an apple to apple comparison.

However, it is still useful to see where the proposed branching strategy stand relative to SCIP and some problem-specific tuned version of SCIP. Specifically, we ran a grid search on 3 parameters of SCIP: (a) Presolve, (b) Heuristics, (c) Separating. Each of these parameters can take four values: (a) OFF - do not use it, (b) DEFAULT: keep it at the default level (SCIP without tuning will use this setting), (c) AGGRESSIVE: use it aggressively, and (d) FAST: use it, but do not spend too much time on it. Based on the Eq. 8 (T=30mins), we found the following parameters suitable for the problem families: (a) cauctions - (AGGRESSIVE, DEFAULT, OFF), (b) setcover: (DEFAULT, DEFAULT, OFF), (c) facilities: (DEFAULT, DEFAULT, FAST), (d) indset: (FAST, OFF, FAST). Table 14 shows the performance comparison with SCIP and TunedSCIP. Specifically, we find that the proposed GNN-PAT retains its performance superiority over the default SCIP as well as the tuned version of SCIP.

Table 14: Performance of *branching strategies* compared with SCIP and TunedSCIP. The best performing numbers are in **bold**. Refer to Section 6 for metrics.

| Model | Time | Time (c) | Wins | Solved | Nodes (c) |
|---|---|---|---|---|---|
| FSB[*] | n/a | n/a | n/a | n/a | n/a |
| RPB | 626.81 | 434.92 | 0 | 80 | 17 979 |
| TUNEDRPB | 644.20 | 450.06 | 0 | 80 | 18 104 |
| SCIP | 582.47 | 396.90 | 4 | 81 | 17 979 |
| TUNEDSCIP | 581.58 | 396.98 | 2 | 82 | 19 937 |
| GNN | 507.06 | 333.59 | 14 | 80 | 17 145 |
| GNN-PAT (ours) | **477.26** | **310.22** | **64** | **84** | **16 388** |
| Combinatorial Auction (Bigger) | | | | | |
| FSB | 2700 | n/a | 0 | 0 | n/a |
| RPB | 1883.32 | 1213.57 | 0 | 47 | 58 766 |
| TUNEDRPB | 1851.83 | 1168.58 | 0 | 48 | 58 155 |
| SCIP | 1755.15 | 1048.76 | 1 | 54 | 61 546 |
| TUNEDSCIP | 1683.81 | 963.77 | 8 | 52 | 60 496 |
| GNN | 1708.99 | 991.07 | 3 | 54 | 39 535 |
| GNN-PAT (ours) | **1601.28** | **892.85** | **48** | **59** | **38 385** |
| Set Covering | | | | | |
| FSB | 917.39 | 758.19 | 7 | 85 | 50 |
| RPB | 737.66 | 607.19 | 1 | 92 | 104 |
| TUNEDRPB | 751.27 | 619.27 | 1 | 91 | 97 |
| SCIP | 677.37 | 565.91 | 10 | **95** | **86** |
| TUNEDSCIP | 656.72 | 538.61 | 18 | 94 | 88 |
| GNN | 646.03 | 525.80 | 14 | 92 | 293 |
| GNN-PAT (ours) | **581.91** | **471.20** | **44** | **95** | 304 |
| Capacitated Facility Location | | | | | |
| FSB | 2700 | n/a | 0 | 0 | n/a |
| RPB | 1984.91 | 865.90 | 0 | 32 | 11 886 |
| TUNEDRPB | 2016.85 | 927.68 | 0 | 33 | 12 486 |
| SCIP | 1920.46 | 784.62 | 0 | 37 | 11 886 |
| TUNEDSCIP | 1578.01 | 443.89 | 2 | 48 | 10 392 |
| GNN | 1207.62 | 262.61 | 14 | 65 | 7233 |
| GNN-PAT (ours) | **1035.32** | **223.81** | **54** | **70** | **5722** |
| Maximum Independent Set | | | | | |

## Q  Results with 10,100-shifted geometric mean of time and nodes

We followed the evaluation protocol proposed by Gasse et al. (2019)(see Evaluation paragraph in Section 5 of Gasse et al. (2019)). Therefore, our results in Tables 2 are based on 1-shifted geometric means of nodes and time. However, since Achterberg (2007) studied a much more diverse set of instances with widely varying solving behavior, they use a 100,10-shifted geometric means respectively. We are not in this setting as our instances come from a specific distribution. Nonetheless, in Table 15 we show the results with 10,100-shifted geometric means. Our conclusions will still remain same.

Table 15: Performance of *branching strategies* on *Big* evaluation instances with 10,100-shifted geometric mean for time and nodes respectively. The best performing numbers are in **bold**. Refer to Section 6 for metrics. Since all of the *Big* Combinatorial Auctions instances were solved to optimality, we extend the evaluation to *Bigger* instances (*since only 10 instances were solved using FSB, we omit it here; See Appendix K for the full results including those on *Big* instances.)

| Model | Time | Time (c) | Wins | Solved | Nodes (c) |
|---|---|---|---|---|---|
| FSB* | n/a | n/a | n/a | n/a | n/a |
| RPB | 633.20 | 438.94 | 1 | 80 | 18 052 |
| TUNEDRPB | 650.24 | 453.86 | 0 | 80 | 18 169 |
| GNN | 515.04 | 338.33 | 14 | 80 | 17 217 |
| GNN-PAT (ours) | **484.45** | **314.17** | **69** | **84** | **16 448** |
| Combinatorial Auction (Bigger) | | | | | |
| FSB | 2700 | n/a | 0 | 0 | n/a |
| RPB | 1884.49 | 1214.20 | 1 | 47 | 58 781 |
| TUNEDRPB | 1853.08 | 1169.20 | 0 | 48 | 58 169 |
| GNN | 1710.74 | 991.84 | 5 | 54 | 39 548 |
| GNN-PAT (ours) | **1603.20** | **893.58** | **54** | **59** | **38 399** |
| Set Covering | | | | | |
| FSB | 917.39 | 758.19 | 8 | 85 | 84 |
| RPB | 740.16 | 608.88 | 3 | 92 | 157 |
| TUNEDRPB | 753.67 | 620.88 | 2 | 91 | **149** |
| GNN | 648.72 | 527.57 | 16 | 92 | 330 |
| GNN-PAT (ours) | **584.60** | **472.93** | **66** | **95** | 339 |
| Capacitated Facility Location | | | | | |
| FSB | 2700 | n/a | 0 | 0 | n/a |
| RPB | 1987.27 | 889.92 | 0 | 32 | 12 443 |
| TUNEDRPB | 2019.12 | 954.35 | 0 | 33 | 12 976 |
| GNN | 1215.48 | 281.93 | 14 | 65 | 8009 |
| GNN-PAT (ours) | **1043.12** | **235.78** | **56** | **70** | **6186** |
| Maximum Independent Set | | | | | |

