# OpenReview forum: "Lookback for Learning to Branch"
_TMLR — Accepted by TMLR_

### Review · Reviewer_X9jE · 2022-07-11

**Summary Of Contributions:**

Summary.

This paper is dedicated to leveraging the "lookback" phenomenon in the B&B solvers. The "lookback" phenomenon means that with the strong branching heuristic, a child node's best choice was often the parent's second-best choice. Specifically, the authors introduce two methods, (1) target smoothing for the standard cross-entropy loss; (2) adding a parent-as-target lookback regularizer. They validate their proposals on standard benchmark instances and show 15% improvements in the solving times.

**Broader Impact Concerns:**

I do not see obvious ethical concerns.

**Requested Changes:**

Refer to the weakness section.

**Strengths And Weaknesses:**

Pros.

1. The paper is well written and easy to follow.

2. The proposed methods are well motivated. First, they use the target smoothing to increase the frequency of the "lookback" phenomenon; second, they add regularization to make use of it.



Cons.

1. The related works are quite old. As shown in Section 2, there is only one paper from 2022 and all other references are from before 2020. The authors need to include and discuss more relevant works arisen in 2021 and 2022.

2. Missing discussions and comparisons with other efficient B&B solvers. Therefore, it is hard to know whether 15% time savings are significant or not.

3. What is the definition of frequency of the loopback property in Figure 1?

4. It seems the lookback phenomenon is not unique for the GNN B&B solver. I am curious about whether the proposed methods can also utilize lookback in other kinds of B&B solvers. If there is any unique relationship between lookback and GNN solvers, the authors need to clarify.

---

> ### Author Response · Authors · 2022-07-25
> **Thank you for time to review our work!**
>
> Dear reviewer,
>
> Thank you for your time in reviewing our work. We are glad that you found our work easy to follow. Here, we provide a point-by-point answer to your concerns.
>
> **1. The related works are quite old….**
>
> Thank you for raising this concern. We redid a thorough literature search and plan to expand our related works section by including three more works. However, due to the character limit in this reply, we have uploaded the paper with the revised related works section.
> We will also be grateful if you can point us to any other work that seems relevant but we haven’t mentioned.
>
> **2. Missing discussions and comparisons with other efficient B&B solvers…**
>
> Our method, as well as the competing machine learning methods, are trained on samples and features collected from SCIP. The models are thus SCIP-specific and it makes sense to compare them against the default rule in SCIP, as is in any case the standard practice in the evaluation of modifications to MIP solvers (Acheterberg et al., 2013). In fact, because the features are somewhat solver-specific, even if we would like to, a SCIP-trained model can not be evaluated on a commercial solver, say, CPLEX or Gurobi, which do not expose the same features as SCIP.
>
> Now, all state-of-the-art solvers work essentially in the same way, with a branch-and-cut backbone - they only differ on implementation details. Thus, we would expect that repeating the same experiments on, say, CPLEX or Gurobi would yield the same amount of improvement: that is, we believe that a CPLEX-trained lookback GNN model on CPLEX-specific features would probably offer something on the order of 15% improvement against default CPLEX.
>
>
> Unfortunately, since these solvers are closed-source, actually implementing our approach in CPLEX or Gurobi would be somewhat difficult for now. The work we base ourselves on, Gasse et al. (2019) had to patch SCIP (for example, to implement a “vanilla” strong branching rule). This is not to say that our method could not be used in principle in CPLEX or Gurobi: rather that it might require some involvement from their development teams. Thankfully, with the growing number of papers demonstrating improvements in solving speed with machine learning techniques, it is increasingly likely that the development teams of these closed-source solvers will be interested in making their software more machine-learning-friendly. In any case, this is merely a temporary political obstacle, not a scientific one.
>
> Finally, within SCIP itself, we should mention that SCIP 8 (Bestuzheva et al., 2021) reports 17% time savings compared to SCIP 7 (see Table 1, first row, second to the last column). This 17% is the result of many new solver features, as discussed in that paper. In contrast, our method can yield comparable running time reductions of up to 15% with only a change to the branching strategy, at least on the datasets we evaluated. Therefore, those gains seem quite significant.
>
> **3. What is the definition of frequency of the loopback property in Figure 1?**
>
> We are sorry that the meaning was not clear: “frequency of the lookback property” refers to the fraction of times the B&B tree formed by the strong branching heuristic exhibited the lookback property between the parent and the child nodes. We will clarify this in the paper.
>
> **4. It seems the lookback phenomenon is not unique for the GNN B&B solver…**
>
> No indeed, there is nothing particularly specific about GNNs in the lookback property: it is a possible phenomenon or characteristic of branching rules in general. In fact, part of our work is to demonstrate that it essentially holds for the strong branching rule, which is not even a machine learning method. GNNs appear in our work only because they are currently the state-of-the-art approach for learning to branch, not because they have any specific relationship with the lookback property.
>
> However, this question brings up different future directions to explore. The work we present regards variable selection, but it is possible that the notion of lookback property could hold (with modifications) for other tasks, such as cutting plane or primal heuristic selection. Although we do not explore this question in this work, we hope our work will serve as a stepping stone toward such explorations.
>
> [References]
>
> Achterrberg et al. (2013) Mixed-Integer Programming: Analyzing 12 Years of Progress (https://link.springer.com/chapter/10.1007/978-3-642-38189-8_18)
>
> Gasse et al. 2019 Exact Combinatorial Optimization with Graph Convolutional Neural Networks (https://arxiv.org/abs/1906.01629)
>
> Bestuzheva et al. 2021 The SCIP Optimization Suite 8.0 (http://www.optimization-online.org/DB_FILE/2021/12/8728.pdf)

---

### Review · Reviewer_KxeK · 2022-07-17

**Summary Of Contributions:**

The paper studies the lookback phenomenon: in the branch and bound tree the strong branching heuristic would often predict a child's best variable as a second best variable one layer above. Authors explore ways of incorporating this phenomenon into GNNs trained to imitate strong branching. Empirical evaluation on 4 synthetic datasets from Gasse et al, 2019 suggests that GNN trained with Parent-as-target (PAT) lookback loss indeed better imitates strong branching and leads to shorter solving time. Authors additionally discuss a model selection framework that specifically assess generalization from smaller MIP to larger MIP instances.

**Broader Impact Concerns:**

No concerns.

**Requested Changes:**

1. Demonstrate the presence of the lookback phenomenon in a diverse set of real-world MIPs. In my opinion, without this data the paper cannot be accepted.
2. Provide the configuration details for SCIP and add SCIP as a baseline. This is also quite crucial to understand where is the proposed method relative to the set of heuristics that are already included in SCIP.

**Strengths And Weaknesses:**

## Strengths

The observed lookback phenomenon looks quite original to me. Finding similar regularities in other branching heuristics may lead to other improvements in ML-driven MIP solvers.

## Weaknesses

### Use of synthetic data only

My biggest concern regarding this paper is the use of synthetic datasets for both getting the inspiration for the lookback phenomenon and evaluation. I cannot be sure that lookback is as clearly observed in a diverse enough set of real-world MIPs and if that is not the case, then this paper's impact is fairly limited. (Nair et al, 2020) can serve as at least an initial source of more realistic datasets.

### Lack of SCIP details

Parameters of SCIP can significantly affect solving time and are in some sense another set of hyperparameters for both training and evaluation. I would expect authors to ideally motivate their choice or at least report the configuration. It could be worth using SCIP (perhaps, with parameters tuned for each problem set individually) as a baseline. Outperforming tuned SCIP can indeed be regarded as a significant contribution to practical MIP solving.

### Minor comments

FSB abbreviature is used in the text but is actually never properly introduced. Does it mean full strong branching?

---

> ### Author Response · Authors · 2022-07-25
> **Thank you for your valuable feedback!**
>
> Dear reviewer,
>
> Thank you for your time. Here, we answer the concerns section by section.
>
> **Real-world MIPs:**
>
> Thank you for raising this valid concern. To address this, we considered two real-world instances - CORLAT (corridor planning in wildlife management) and RCW (Red-cockaded woodpecker diffusion conservation instances ). Note that CORLAT was also used by Nair et al. 2020.
> We found that the lookback property occurs 34.73% of the times in CORLAT instances and 38.16% of the times in RCW instances. We have added this discussion in Appendix A of the newly uploaded paper.
>
> **SCIP details:**
>
> To provide accurate comparisons with our main competitor, Gasse et al. (2019), we decided to set SCIP the same way as they do, which can be found at this link: https://github.com/ds4dm/learn2branch/blob/master/utilities.py#L16. Specifically, in their work SCIP is used with default parameters, except that:
> Cutting planes are allowed only at the root node;
> Restarts are disabled.
>
> We, therefore, did the same. Thank you for pointing it out - we will clarify this in the paper (see Appendix H).
>
> **Comparison with SCIP and TunedSCIP:**
>
> As explained above, SCIP, by default uses lots of heuristics. As per the standard practice to evaluate branching strategies, these heuristics are turned down. However, with this caveat in mind, we have produced a comparison table in Appendix P of the newly uploaded paper. This comparison is with SCIP and the tuned version of  SCIP as proposed by Nair et al. 2020. In Appendix P, we provide more details about tuning, but in summary, our proposed GNN-PAT outperforms both the default SCIP and its tuned version.
>
> **FSB:**
>
> Thank you for pointing this out. Yes. FSB is the full strong branching heuristic.

---

> > ### Comment · Reviewer_KxeK · 2022-07-30
> > **Reply**
> >
> > Thank you for your reply. I have acknowledged the changes and am satisfied with them.

---

### Review · Reviewer_FN7m · 2022-08-19

**Summary Of Contributions:**

The paper extends the work of Gasse et al on using GNNs to imitate strong branching, the strongest heuristic we know in terms of number of nodes, in order to solve instances of MIPs drawn from the same problem. The work is based on the observation that the second-best variables to branch on at a node according to strong branching is often the best to branch on at the children of that node. The authors show that Gasse’s GNN does not exhibit this property to the same extent as strong branching, and therefore posit that a GNN that would exhibit this property would be a closer, stronger imitation of strong branching. Thus, the paper defines loss functions to encourage this behavior. Experiments show that the performance of the resulting branching rule is significantly improved.

**Broader Impact Concerns:**

No concerns.

**Requested Changes:**

MAJOR
The paper uses shifted geometric mean with a shift of 1 for both time and nodes. However, the standard is 10 and 100, respectively, including in Tobias Achterberg’s PhD thesis, referenced by this paper. It would be good if the authors could justify why this standard was not followed, or follow it.

MINOR
There are four minors issues with the last paragraph of Section 3.
First, using - to denote the child obtained by adding a lower bound to the branching candidate is counterintuitive, because adding a lower bound will increase the value of the variable in the child’s LP.
Second, the argmax is taken over each branching candidates i, but the index i does not appear in the argmax, so it is unclear what changes in this expression when i changes. I would add i as a subscript to P^- and P^+.
Third, I’d add epsilon_LP > 0 to be clear.
Four, I’d add a reference to Tobias Achterberg’s PhD thesis for this section (already cited elsewhere in the paper).

In paragraph 2 of Section 4, I find the notation confusing. In Section 3, the letter i is used to index branching candidates. At the beginning of the first sentence of paragraph 2 of Section 4, it sounds like instances are indexed by i, but at the end of the sentence, we are told that nodes are indexed by i. Later in that paragraph, j is used to index variables. A small re-write would be good.

In Section 4.1, “equally distributed” is not formal. Furthermore, I suspect that the choice of parameter epsilon will be discussed later in the paper, but it might be good to say so at this point. “is still not aware of their parent behavior” is informal. I think I get the intuition, but I’m not sure.

Section 5, paragraph 3: in my experience, practitioners do not care about criterion (c) Minimum node count. I would either further justify its inclusion, or remove it.

Equation (7) is not clear. In particular, why is the number of nodes n() involved in selecting the branching strategy with the best solving time? Furthermore, at this point of the read, it is not clear to me why we need equation (7) at all.


**Strengths And Weaknesses:**

The paper is pretty clear and well presented. The logic flows well. The observation that Gasse’s GNN does not exhibit the “second-best” property of strong branching to the same extent is interesting. The authors modify the GNN to better imitate strong branching, which leads to improvements.

In my mind, the biggest weakness of the paper is in the way that it uses the observation that good branching candidates at one node are often good branching candidates at its children. Currently, I would describe the approach as a “patch”. It “just” gives an incentive to pick good branching candidates at the parent. Instead, I would be interested in understanding the shortcoming of Gasse’s GNN that makes it unable to exhibit this property naturally (supposing it is a desirable property). Once this shortcoming is identified, it ‘d be possible to change the approach so that the property naturally arises. I think that’d be more interesting and promising.

---

### Decision · Action_Editors · 2022-09-26

**Recommendation:** Accept as is

**Comment:**

The paper studies the lookback phenomenon in machine learning based mixed integer linear programming approaches. The authors point out that the standard GNN based approach does not show strong branching and suggests fixes to imitate strong branching.

Reviewers commonly pointed out that the paper is clear and easy to follow. Also reviewers assessed that the claims made in the paper are accurate and well supported by the experiments. Moreover they pointed out that the proposed method is well motivated and interesting to the wider ML community.

Authors have addressed major concerns (along with multiple minor changes) raised by the reviewers by including real world MIP results (raised by reviewer `KxeK`), expanding evaluation protocol (suggested by reviewer `FN7m`)  and explaining comparison against other efficient B&B solvers (by reviewer `X9jE`). In the end, all reviewers were satisfied with the author's response and changes and supported acceptance.

In the final recommendation, reviewer `KxeK` mentioned: "it's worth accepting the paper to let researchers build on top of it, hopefully proving that these positive results also generalize to larger-scale problems." also reviewer `FN7m` mentioned: "the paper would be interesting to a sizeable part of the ML community."